# The Effect of Particle Necks on the Mechanical Properties of Aerogels

**DOI:** 10.3390/ma16010230

**Published:** 2022-12-27

**Authors:** Lorenz Ratke, Ameya Rege, Shivangi Aney

**Affiliations:** 1German Aerospace Center, Institute of Materials Research, 51147 Cologne, Germany; 2School of Computer Science and Mathematics, Keele University, Staffordshire ST5 5BG, UK

**Keywords:** aerogels, microstructure, non-linear beam bending, corrugations, scaling laws

## Abstract

Mechanical properties of open-porous materials are often described by constructing a cellular network with beams of constant cross sections as the struts of the cells. Such models have been applied to describe, for example, thermal and mechanical properties of aerogels. However, in many aerogels, the pore walls or the skeletal network is better described as a pearl-necklace, in which the particles making up the network appear as a string of pearls. In this paper, we investigate the effect of neck sizes on the mechanical properties of such pore walls. We present an analytical and a numerical solution by modeling these walls as corrugated beams and study the subsequent deviations from the classical scaling theory. Additionally, a full numerical model of such pearl-necklace-like walls with concave necks of varying sizes are simulated. The results of the numerical model are shown to be in good agreement with those resulting from the computational one.

## 1. Introduction

Gibson and Ashby derived scaling laws for three-dimensional (3D) open-porous cellular materials in their seminal book on porous solids [1]. They showed that for many properties, particularly mechanical ones, the relative density determines the property. Here, the relative density is defined as the density of the porous material divided by that of the pore-free material. For instance, Young’s modulus *E* should vary with the relative density in a simple power law fashion $E\propto \rho^{2}$ and the plastic rupture strength $\sigma_{cr}$ should depend on the density in a similar way, namely $\sigma_{cr}\propto \rho^{3/2}$. While recent investigations have asserted the importance of the pore-size distributions and pore-wall morphology, alongside the relative density, in dictating the mechanical properties of such materials [2], the scaling relations still hold true for a vast majority of open-porous materials. This is a consequence of the models proposed by Gibson and Ashby, which work fine because many porous materials exhibit a cellular structure with walls of constant cross section, and in open-porous materials the edges of the pores, the struts, and beams can be described as smooth quadratic bars or cylinders. The pore sizes are assumed to be constant and not exhibiting a size distribution. In aerogels, the pore sizes range from a few ten nanometers to even micrometers, and the struts are, for instance, in the case of silica and Resorcinol–Formaldehyde (RF)-aerogels, better described as a pearl-necklace structure as shown with a lot of scanning electron micrograph (SEM) pictures in the book of Ratke and Gurikov [3]. While recent models have tackled the subject of pore-size or cell-size distributions in such materials [2,4], the effect of network connectivity and the effect of this pearl-necklace-like morphology remains less addressed.

Some newer models described the mechanical properties of aerogels with two- dimensional (2D) or 3D pore model of constant cross-section [5,6,7] taking also into account network defects, like dead ends, dangling beams, and struts. More details on different modeling methodologies for describing aerogels can be found in the perspective by Rege [8]. One such model was proposed by Lei and Liu [6] to characterize silica aerogels. They used cubic cells with struts of constant cross. The simulated loading lead for Young’s modulus to a scaling exponent with envelope density of m=2.04, being close to the value predicted by Gibson and Ashby. In aerogels, not the dead-ends or a lack of their presence dictate a deviation from the Gibson and Ashby exponent of m=2, but it is rather the random connectivity and the effect of the string-of-pearl morphology. Gelb [9] and Abdusalamov [10] have investigated the effect of the connectivity on the mechanical properties. For a more detailed discussion see also the book of Ratke and Gurikov [3]. The effect of the pearl-necklace-like morphology was only recently attacked by the authors [11]. Only few authors investigated the effect of inter-particle necks on the mechanical properties of porous materials. Chen et al. [12] studied the elastic properties of porous ceramic films and showed that the inter-particle necks and their coarsening are decisive for the mechanical properties. In colloidal crystals or more general colloidal assemblies interparticle-necks play an important role, since typically there are only weak bonds between the almost spherical particles touching each other. Dargazany and Itskov [13,14] studied the mechanical behavior of interconnected arrays of colloidal particles assuming that at the touching points the bonds can be described as non-linear springs. In many aerogels, the network of particles consist of moderately up to heavily connected bonds and therefore the Gibson and Ashby [1] model has been applied for aerogels [5,15]. All studies performed so far have shown that the pore collapse occurs as a result of a critical stress arising from buckling or bending of the pore walls or better the connected struts. This leads to subsequent failure and crack formation. Since the kinematics of the deformation modes are dependent on the area moment of inertia of the pore walls, the effect of particle necks or more generally any shape variation that may be described as corrugated beams must be significant. It was recently shown by the authors, that modeling these struts or beams with a constant cross-section may result in too soon of a prediction of failure due to stretching and bending, while a delayed prediction of pore collapse is due to buckling [11]. In this study, the interparticle necks were simplified by modeling overlapping spheres. While this may be acceptable for qualitatively describing their effect, a quantitative description demands more accurate modeling of these interparticle necks. This paper attempts to describe and analyze the mechanical behavior of the pearl-necklace-like pore-walls and more generally corrugated beams as against the one with a constant cross-section and shows how the wall thickness variations modify the scaling laws derived by Gibson and Ashby.

## 2. Description of Pore Walls as Corrugated Beams

Particles form in a gel solution by different mechanisms resulting in the formation of the porous network of silica, resorcinol–formaldehyde (RF) and other polymeric aerogels. For instance, polycondensation reactions lead to entangled polymers, which may be regarded as particles or the particles form via phase separation (nucleation, spinodal decomposition), while the state point of the polymerizing system passes a stable or metastable miscibility gap [3,16]. Once the particles touch, they establish bonds and build particle clusters by various aggregation mechanisms. The connected particles further grow, since the surrounding solvent is still rich in monomers and oligomers. At the circular contact line surrounding the area of contact, they preferably condense since the effect of concave curvature increases the condensation or attachment rate. This growth leads to necks between the particles. Figure 1 shows an SEM of an RF-aerogel clearly depicting a pearl-necklace-like structure. The almost spherical particles with different diameter are connected by necks with a concave curvature.

After gelation and drying, the aerogel microstructure looks as if the particles would overlap, the spheres would have been penetrated into each other and the necks have a concave curvature. In a simplified manner, one can then describe the network of particles as an arrangement of simple cubic boxes, whose edges consists of spheres with radius *R*. If these spheres overlap by a small amount ξ=h/R, meaning they interpenetrate by an amount of h≤R this overlap affects the porosity, the solid fraction and the mechanical properties as recently shown in [11,17]. The elastic and plastic response of the full box or the whole network can be analyzed by considering the mechanical behavior of a beam (strut) located at the box edges. Figure 2 shows a box made with cylindrical edges compared to a box having strings of overlapping pearls at the edges.

However, these spherical particles overlapping in Figure 2 do not consider the development of necks with a concave curvature. Thus, for further calculations, we use another description, which also allows to mimic the interparticle necks. We can describe the geometry of such an array of spherical particles with necks as a rod with cosinusoidal corrugations shown schematically in Figure 3. Then, the radius varies in one direction like
(1)$$R\left(x\right)=R_{0}\left(1-\alpha \cos^{2}\left(\frac{n\pi }{L}x\right)\right)$$
with *L* the length of the rod, *n* the number of wave maxima, and $\alpha =\Delta R/R_{0}$, the amplitude of the corrugations. Formally, 0≤α≤1. We will use later Equation (Equation 1) to calculate the bending of corrugated beams and compare the results with a full finite element calculation of a string-of-pearls with necks, see Section 5.

## 3. A Simple Estimate of Buckling Strength

Gibson and Ashby discuss several failure modes of open-porous foams. The first is the elastic failure, occurring, once the critical buckling load of a strut or beam making up the pore wall of the foam is passed. In their model, they construct the foam as being made up by rectangular beams building a cube of edge length *ℓ* and the beams have a thickness *t*. The beams buckle once the critical load is reached. According to the linear Euler buckling theory, a critical force
(2)$$F_{c}=\frac{\eta^{2}\pi^{2}E_{s}I}{\ell^{2}}$$
must be overcome to buckle a beam. In this expression, η is a factor, which accounts for the type of constraints at the beam end (order 1), $E_{s}$ is Young’s modulus of the beam material, *I* is the second moment of the beam area. The critical stress is then calculated as
(3)$$\sigma_{c}\propto \frac{F_{c}}{\ell^{2}}\propto \frac{E_{s}I}{\ell^{4}}.$$

The second moment of the area is $I\propto t^{4}$ and since the volume fraction of solid in a unit cube of edge length *ℓ* is given as $\phi_{s}\propto {\left(t/\ell \right)}^{2}$ we have the scaling relation for elastic failure of a foam as
(4)$$\frac{\sigma_{c}^{el}}{E_{s}}\propto \phi_{s}^{2}.$$

As described in the previous sections, not all particulate aerogels can be described by simple overlapping spheres, since touching particles develop necks at their point of contact with a radius of curvature defined by several growth mechanisms as sketched above. In principle we could take a unit cell with pore walls like an array of particles with interparticle necks and load it at the top and ask, how they bend. Figure 4 shows a sketch of a box made with spherical particles and a string of spheres with necks is taken out exhibiting a simulated deformation. The forces at the top and bottom of the box lead to a bending of the string of pearls at each edge. The spherical particles will resist such a deformation and stresses will develop at their necks, much larger than the stress in a full circular bar. The necks act as local stress raisers, which one could describe by a notch factor.

For mathematical modeling of the neck between two particles, consider the schematic illustrated in Figure 5. From the sketch, one can derive the following relation for the neck radius,
(5)$${\left(R+\rho \right)}^{2}={\left(x+\rho \right)}^{2}+R^{2}$$
leading to
(6)$$\rho =\frac{x^{2}}{R(1+\frac{x}{R})}.$$

We now calculate the bending of such a string of connected pearls using a notch factor. The notch factor can be expressed by the following relation
(7)$$K_{n}=1+2\sqrt{\frac{c}{\rho }}.$$

This equation would, for instance, give a circular hole punched into a sheet, c=ρ, a notch factor of 3, which is exact for such a case. For a very small neck radius of curvature, the notch factor would be very large and one would expect, that the failure stress is much lower, compared with a circular bar of radius *x*. The second moment of area is now
(8)$$I\propto x^{4}.$$

With the same calculations as above, we come to an equation of the buckling strength
(9)$$\sigma_{c}\propto \phi_{s}^{2}/K_{n}.$$

This is not exact since the solid fraction is only proportional to $x^{2}/\ell^{2}$. The proportionality factor depends on the chosen network model. At this point, we will first neglect this and instead take a closer look at the notch factor. The notch depth *c* can be calculated easily, x+c=R, and from the given geometry we obtain for x≪R
(10)$$K_{n}\approx 1+\frac{R}{x}.$$

Therefore, we finally have
(11)$$\sigma_{c}\propto \frac{\phi_{s}^{2}}{1+\frac{R}{x}}.$$

Making a series expansion up to second order in *x* around x=0, and inserting x/R=1−α, see Equation (Equation 1), leads to
(12)$$\sigma_{c}\propto \phi_{s}^{2}\left(1-\alpha +\alpha^{2}\right).$$

The larger the amplitude of corrugations, the smaller the buckling strength.

## 4. Bending of Corrugated Beams

In the previous section, we neglected that a varying cross-section of a chain of spheres might change the Euler buckling analysis. In the following, we try to calculate the Euler buckling of a chain of spheres. The general equation describing bending of a rod with varying second moment of area I(x) is according to Landau and Lifshitz [18]
(13)$$\frac{d^{2}}{dx^{2}}\left(I\left(x\right)\frac{d^{2}u}{dx^{2}}\right)+\frac{P}{E}\frac{d^{2}u}{dx^{2}}=0.$$

In this equation, u(x) is the displacement perpendicular to the rod axis, *P* is the load acting at the ends of the rod, and *E* is the Young’s modulus. Integrating two times leads to
(14)$$I\left(x\right)\frac{d^{2}u}{dx^{2}}+\frac{P}{E}u=C_{1}x+C_{0},$$
where $C_{1},C_{0}$ are the two constants of integration. Writing u(x)=w(x)+ax+b and inserting leads to a simple equation for the displacement w(x)
(15)$$I\left(x\right)\frac{d^{2}w}{dx^{2}}+\frac{P}{E}w=0,$$
with $a=E/PC_{1}$ and $b=E/PC_{0}$. We aim to solve Equation (Equation 15) for buckling of a string of pearls. For a rod with radius R(x) the second moment of area is
(16)$$I\left(x\right)=\frac{R{\left(x\right)}^{4}}{4}.$$

### 4.1. Approximate Analytical Solution

Insertion of Equation (Equation 1) into Equation (Equation 16) leads to a new expression for the second moment of area as
(17)$$I\left(x\right)=\frac{R_{0}^{4}}{4}{\left(1-\alpha \cos^{2}\left(\frac{n\pi }{L}x\right)\right)}^{4}=I_{0}{\left(1-\alpha \cos^{2}\left(\frac{n\pi }{L}x\right)\right)}^{4}.$$

Insertion into Equation (Equation 15) unfortunately leads to an equation that cannot be solved analytically. We therefore rewrite this equation as
(18)$$\frac{d^{2}w}{dx^{2}}+\frac{P}{EI(x)}w=0.$$

Insertion of the full expression of the second moment of inertia still would lead to an equation that cannot be solved analytically. However, a series expansion of the inverse of the second moment of inertia to second order yields the following equation
(19)$$\frac{d^{2}w}{dx^{2}}+\frac{P}{EI_{0}}\left(1+4\alpha \cos^{2}\left(\frac{n\pi }{L}x\right)\right)w=0.$$

This differential equation can be solved directly. The solution fulfilling the boundary conditions w(0)=0 and w(L)=0 are Mathieu functions of odd parity, named here $M_{s}\left(a,q,x\right)$, with a,q suitable constants (One can imagine this using the identity $\cos^{2}\left(x\right)=1/2\left(1+\cos\left(2x\right)\right)$, which transforms the square of the cosine and then one looks into textbooks on differential equations to see that the resulting equation is Mathieu’s equation. For details on Mathieu functions, see [19,20]).
(20)$$w\left(x\right)=w_{0}M_{s}\left(\frac{4L^{2}P\left(1+2\alpha \right)}{En^{2}\pi^{2}I_{0}},-\frac{L^{2}P\alpha }{EI_{0}n^{2}\pi^{2}},\frac{n\pi }{L}x\right).$$

In the case of a rod without corrugations, we have α=0 and then the Mathieu function becomes a simple sinus
(21)$$w\left(x\right)=w_{0}\sin\left(\frac{n}{L}\sqrt{\frac{L^{2}P}{EI_{0}n^{2}}}x\right).$$

Using the boundary condition at x=L leads to
(22)$$\sin\left(\sqrt{\frac{L^{2}P}{EI_{0}}}x\right)=0$$

The sine is zero at all multiple of π, namely at mπ, m=0,1,2,3,... and thus, we get the Euler criterion for buckling
(23)$$P_{0}=\frac{m^{2}\pi^{2}EI_{0}}{L^{2}}=\frac{m^{2}\pi^{2}ER_{0}^{4}}{4L^{2}}.$$

The simplest mode of buckling is at m=1. At least for vanishing corrugations, the result is in agreement with the simple solution of the bending equation. We can make use of this result to simplify Equation (Equation 20) to
(24)$$w\left(x\right)=w_{0}M_{s}\left(\frac{4(1+2\alpha )}{n^{2}}\frac{P}{P_{0}},-\frac{\alpha }{n^{2}}\frac{P}{P_{0}},\frac{n\pi }{L}x\right).$$

At the boundary x=L, we should have w(L)=0, and thus
(25)$$M_{s}\left(\frac{4(1+2\alpha )}{n^{2}}\frac{P}{P_{0}},-\frac{\alpha }{n^{2}}\frac{P}{P_{0}},n\pi \right)=0.$$

The solution of this equation yields the buckling mode of a corrugated rod. Unfortunately there is no analytical expression for the zeros of Mathieu’s function available. Defining $\chi =P/P_{0}$, we can solve Equation (Equation 25) numerically looking for the relative buckling load χ as a function of α at fixed *n* for instance. A result of such a calculation is shown in Figure 8 as the upper dashed line with full circles using n=10 in Equation (Equation 1). Although the result shows the expected trend, namely a reduction of the buckling load with increasing amplitude of the corrugations, one should not overstress the result. The series expansion of the inverse second moment of area made above is only valid until approximately α≤0.1. For larger corrugations or smaller neck areas, one has to solve the bending of a string of overlapping pearls numerically.

### 4.2. Nonlinear Bending-Numerical Solution

Under non-linear bending, Equation (Equation 15) writes
(26)$$\frac{I(x)}{{\left(1+w^{\prime }{\left(x\right)}^{2}\right)}^{3/2}}w^{\prime \prime }+\frac{P}{E}w=0.$$

Before continuing, we define the dimensionless variables: ξ=x/L as a dimensionless coordinate, w=w/L as a dimensionless deflection. Again using the expression for the second moment of area given in Equation (Equation 1), we obtain a new differential equation
(27)$$\frac{{\left(1-\alpha \cos^{2}\left(n\pi \xi \right)\right)}^{4}}{{\left(1+w_{\xi }^{2}\right)}^{3/2}}w_{\xi \xi }+k_{c}w\left(\xi \right)=0,$$
with
(28)$$k_{c}=\frac{PL^{2}}{EI_{0}}\;\mathrm{and}\;w_{\xi \xi }=\frac{d^{2}w}{d\xi^{2}}.$$

The boundary conditions read w(0)=w(1)=0. We solve this equation with an implicit Runge–Kutta method using a Mathematica™notebook. In the numerical solution, interestingly, one only gets bending if the value of $k_{c}$ is above a threshold value. Below this value, the numerical solutions are always close to zero and if the value of $k_{c}$ is much larger the curve often has more than one extreme. Determining the smallest value of $k_{c}$ (threshold) for which we get a simple bending curve, should be the smallest possible value of the critical load $P_{c}$ needed to buckle the beam, given that all other parameters $L,E,I_{0}$ are constants. We always used a value of n=10. At larger values of α, beyond 0.7, no stable solution could be obtained. Higher values of α do not make sense, because at very small necks the bonding between the particles becomes important and this leads to new effects [21,22]. The full line with open circles in Figure 8 shows the results of the numerical solution of Equation (Equation 27) using for the variation of the beam radius expression as given in Equation (Equation 1). The relative buckling load, $P_{c}/P_{0}$ decreases with the amplitude of the corrugations α. $P_{0}$ is the critical load of the α=0-case.

The critical load decreases rapidly with the amplitude and a comparison with the approximate analytical solution of the linear bending equation shows a good agreement up to α=0.1. Let us now compare the result with the solution given in Equation (Equation 11). The amplitude of the corrugations relates to the relative neck radius such as
(29)α=1−x/R,
since at α=0 there are no corrugations and the neck radius equals the particles radius. At α=1, the neck radius is zero. Inserting into Equation (Equation 11) and dividing the result by the expression for the buckling strength as noted in Equation (Equation 4) yields
(30)$$\frac{\sigma_{c}}{\sigma_{unc}}\approx 1-\alpha ,$$
where $\sigma_{unc}$ is the strength of an uncorrugated beam as derived by Gibson and Ashby. Performing a least square fit to the numerical results shown in Figure 8 yields an interesting result. The data can be fitted almost perfectly by the following quadratic equation for α
(31)$$\frac{P_{c}}{P_{0}}=1-2.15579\alpha +1.106\alpha^{2}.$$

Besides the quadratic term, the numerical solution fits very well to the simple estimate given above with the notch factor (there is the important difference, that the nonlinear bending gives not simply the expression 1−α but the decrease of buckling strength is greater by a factor of around 2). It also shows, that the scaling solution, meaning the proportionality to the square of the fraction solid, is not modified, just the prefactor is much smaller and depends on the amplitude of the corrugations. The thinner the necks, the smaller the buckling strength.

## 5. Finite Element Modeling of the Pore Walls with Particle Necks

In our previous study [11], a finite element model of pore walls modeled with overlapping spheres representing particles with necks was described. While overlapping spheres appear like particles having necks, these are, strictly speaking, not an accurate description. Particle necks can be more accurately represented by the geometry shown in Figure 5. In this section, we use this approach for computationally designing model pore walls with varying neck sizes (x/R). Figure 6 exhibits these model pore walls with x/R∈[0.2,1.0] in the reference state. An x/R of 0.2 shows very small necks, while increasing this factor results in wider necks ultimately resulting in a fiber-like array of particles having a constant cross-section. The illustrations shown in Figure 6 give an accurate description of evolving neck sizes in aerogels. These models were generated in ABAQUS and simulated for their buckling behavior using a perturbation analysis. We used a linear elastic model, in which the solid material has a Young’s modulus of 1.5 GPa and a Poisson ratio of 0.3. Solid element type C3D8R was used, which is a general purpose brick element with reduced integration. This element type avoids the phenomena.

The results from the buckling analysis are illustrated in Figure 7. While visually, the deflection of all the pore walls appears similar, quantifying the relative critical buckling load $P_{c}/P_{0}$ for each case shows significant deviations. This $P_{c}/P_{0}$ for varying neck sizes x/R represented as α (see Equation (Equation 29)) are plotted in Figure 8. The numerical solution from Equation (Equation 27) agrees well with these computational results. The small difference between the computational results and the FEM data are probably a result of the geometrical difference between a beam corrugated by a cosinusoidal function and the more realistic shape used in the FEM calculations. In both cases, it was observed that the smaller the neck sizes, the higher the stress-concentration at the necks and the smaller is the buckling strength. This result can be applied to understand the fragility in aerogels and subsequent quantification of their mechanical strength upon, e.g., aging. Mechanical strengthening of silica aerogels by aging was investigated by Einarsrud et al. [23,24]. They showed that with increasing aging time, the strength increases as a result of neck thickening. Unfortunately, they did not measure the neck radius and the particle radius explicitly. Moreover, one can observe that $P_{c}/P_{0}\left(\alpha \right)$ for different particle radii coalesce. This shows that the critical load is independent of the particle size.

## 6. Conclusions

In this paper, an accurate mathematical and computational description of the pearl-necklace-like pore walls with interparticle necks, as observed in aerogels, is presented. The pore walls are modeled as corrugated beams and their bending and buckling solutions, including their nonlinear analysis, are presented. These numerical solutions are then validated by presenting a full finite element calculation of such pore walls with varying interparticle necks. It was observed that the smaller the interparticle neck sizes, the higher the stress-concentration at the necks and smaller the buckling strength. This poses a possible explanation for the fragility in silica or RF aerogels.

## Figures and Tables

**Figure 1 materials-16-00230-f001:**
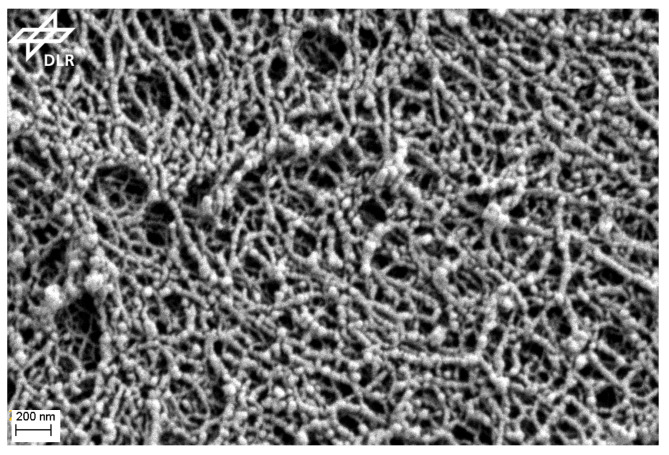
SEM picture of an RF-aerogel with a pearl-necklace microstructure. Picture provided by M. Schwan, DLR.

**Figure 2 materials-16-00230-f002:**
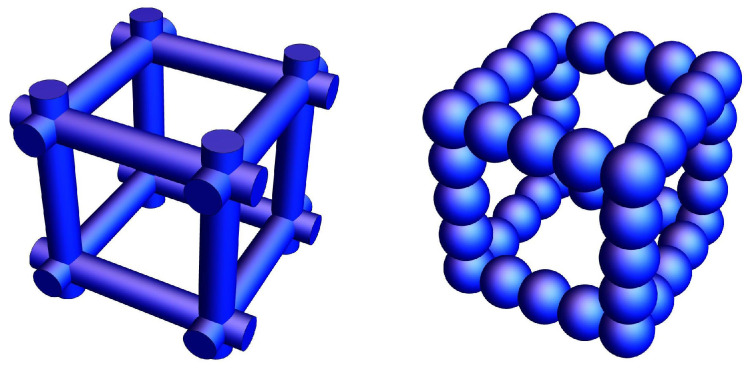
Scheme of possible edges of simple cubic boxes making a porous body. The cylinder in the left upper corner is replaced by string of spheres with increasing overlap to the right bottom corner.

**Figure 3 materials-16-00230-f003:**
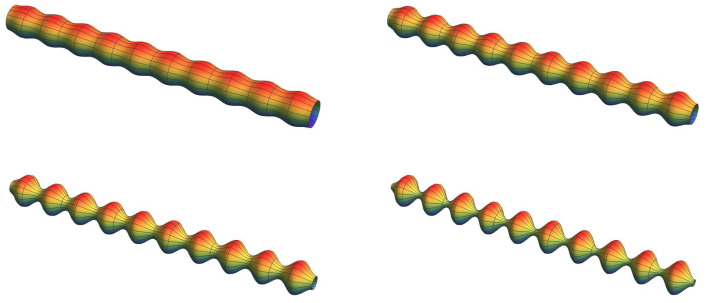
Surface of revolution of a rod with a cosinusoidal corrugations mimicking a chain of spheres connected by necks. In this figure, $R_{0}$ was set to 0.05 and n=10. From upper left to lower right, α varies from 0.2 to 0.4, to 0.6 to 0.8.

**Figure 4 materials-16-00230-f004:**
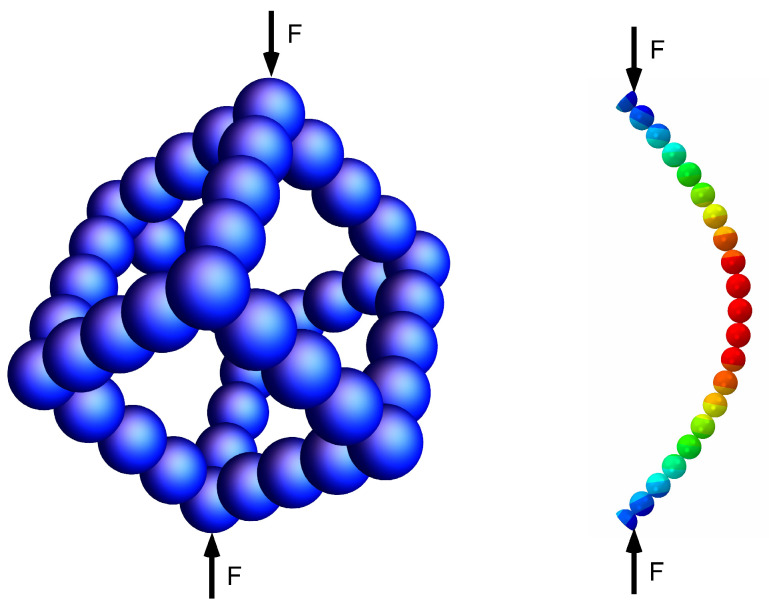
Scheme of a box made with overlapping spheres rotated such that at the top and bottom they can be loaded with a force *F*. This leads to a bending of the string of pearls at each edge, which is shown in the right figure (a finite element simulation discussed in Section 5). The spherical particles will resist such a deformation and stresses will develop at their necks. The necks act as local stress risers, which can be described by a notch factor.

**Figure 5 materials-16-00230-f005:**
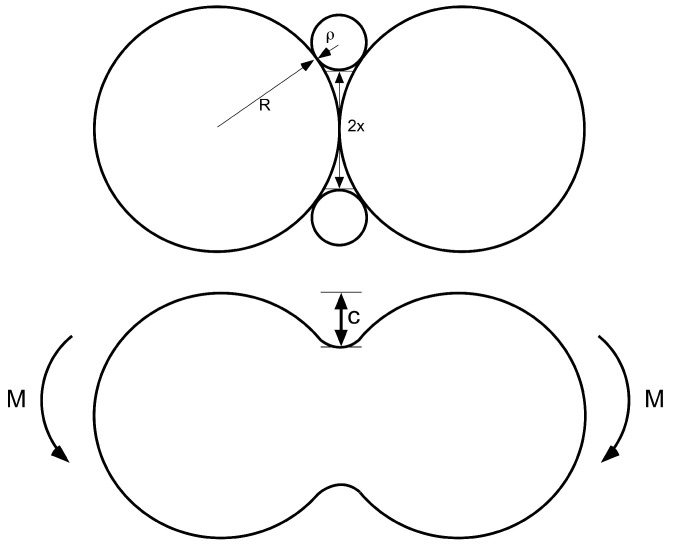
Scheme of two overlapping particles, which built a neck with a radius of curvature ρ during growth. The neck diameter, which can be measured for instance from SEM pictures (see [3]), is denoted as 2x.

**Figure 6 materials-16-00230-f006:**
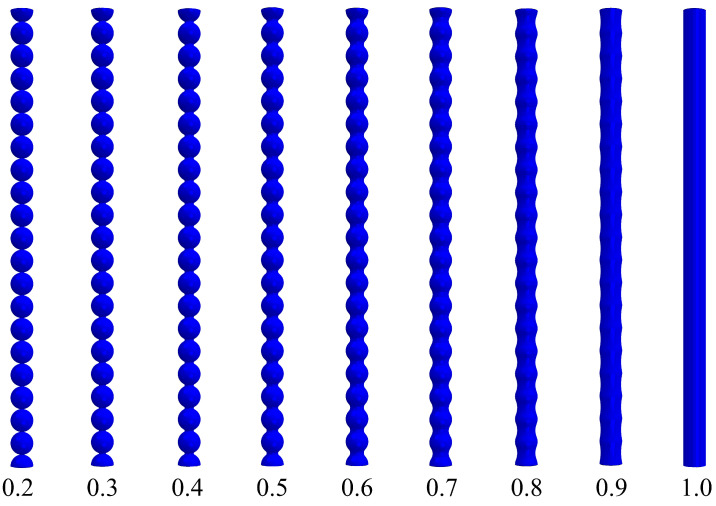
Computational models of the model pore walls in the reference state for different x/R ratios.

**Figure 7 materials-16-00230-f007:**
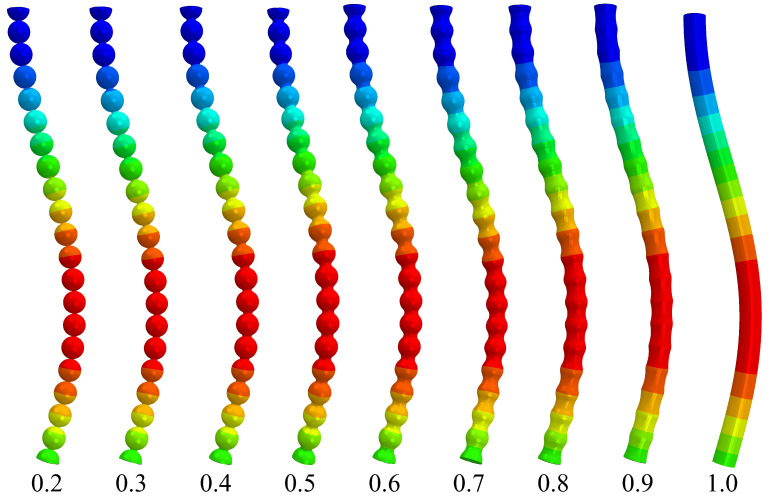
Finite element simulation results for the displacement states of the model pore walls in the deformed state under buckling for different ratios of neck size *x* to particle radius *R*, x/R.

**Figure 8 materials-16-00230-f008:**
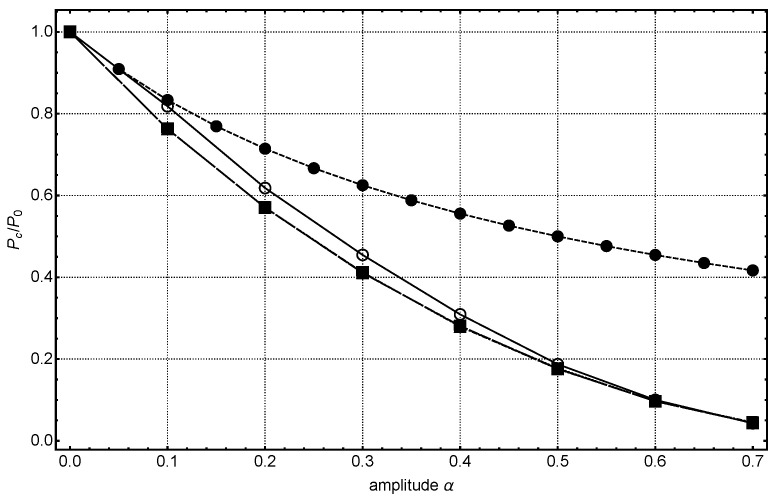
Critical buckling load as it depends on the amplitude of corrugations or the neck size. The full line with open circles is the result of the corrugated beam, see Equation (Equation 27). The dashed line with the full circles is the analytical result of the linear Euler theory, see Equation (Equation 25) and the long-dashed line with the full squares presents the result of the FEM calculations of a string of pearls with concave necks. This curve is also the result for three different particle radii. They all coalesce and thus only the value of α is important.

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
