# Peer review of "The Effect of Particle Necks on the Mechanical Properties of Aerogels"

_materials, 2022, doi:10.3390/ma16010230_

Round 1
Reviewer 1 Report
I read the manuscript from Ratke et al with high interests. The paper investigate the effect of neck sizes on the mechanical properties of pearl-necklace like pore walls. This paper attempts to describe and analyse the mechanical behavior of the pearl-necklace-like pore-walls and more generally corrugated beams as against the one with a constant cross-section and shows how the wall thickness variations modify the scaling laws derived by Gibson and Ashby. Some Comments are listed as follow:
1. Abbreviations should be interpreted for the first time, such as RF.
2. The scale bar in Figure 1 is indistinct. I hope the author can provide SEM images with different magnification。
3. The scale bar of Figure 8 and 9 is missing.

Author Response
1. Abbreviations should be interpreted for the first time, such as RF.
done
2. The scale bar in Figure 1 is indistinct. I hope the author can provide SEM images with different magnification。
Imroved! Thanks for the hint
3. The scale bar of Figure 8 and 9 is missing.
There are no scale bars, because they do not make any sense. The essential point in both figure is the ratio of the neck size x to the particle radius R. Changed the text a bit to make that clear.
Reviewer 2 Report
The manuscript "materials-2072035" by Ratke et al. reported The effect of particle necks on the mechanical properties of aerogels. After review, this study is interesting and I enjoyed reading it. But it remains only a minor correction.
a. Correct reference format according to journal.
Author Response
Tanks for the hint, but the reference format is done automatically using bibtex entries and the style sheet provided by mdpi.
Reviewer 3 Report
Aerogels and solid foams are an interesting class of materials, the understanding of whose properties is still quite young. In this article, some strength properties of such materials, where the pore wall is a corrugated beam, are theoretically and numerically investigated. When constructing the model, the authors rely on images of such walls from an electron microscope. As a result, they present an analytical solution for the critical load of a corrugated beam, which performs well in the finite element computational test. It is shown that the smaller the dimensions of the interparticle necks, the lower the resistance to buckling. In my opinion, the analytically solved problem reflects the deepest layer of our understanding of the nature of materials, so the results are worthy of publication in the Materials. I would like to give just a couple of recommendations.
Images of corrugated beams are truly mesmerizing. However, fig. 3 and 8 look like they are repeated. It may be enough to leave just one of them. The same applies to Fig.6,7 and 10. Fig.10 contains all the information of Fig.6,7. Probably enough to leave only this Fig.10.
Author Response
Images of corrugated beams are truly mesmerizing. However, fig. 3 and 8 look like they are repeated.
My answer: No, they are not repeated. Figure 3 describes beams corrugated with the cosinusoidal corrugations and fig 8 is based on the two particle model shown in figure 5.
It may be enough to leave just one of them. The same applies to Fig.6,7 and 10. Fig.10 contains all the information of Fig.6,7. Probably enough to leave only this Fig.10.
Answer: you made a good point, since indeed all results are in figure 10. After thinking about it, I dropped figures 6 and 7 and only left figure 10.